# Physical activity and sedentary behaviour interventions for people living with both frailty and multiple long-term conditions: a scoping review protocol

Hannah M L Young [ID],[1,2] Thomas Yates,[2] Paddy C Dempsey,[2,3,4] Louisa Y Herring,[1] Joseph Henson,[1,2] Jack Sargeant,[1,2] Ffion Curtis,[5] Harini Sathanapally,[5] Patrick J Highton,[5] Michelle Hadjiconstantinou,[2] Rebecca Pritchard [ID],[6,7] Selina Lock,[8] Sally J Singh,[9,10] Melanie J Davies[1,2]

For numbered affiliations see end of article.

**Correspondence to**
Dr Hannah M L Young;
hy162@le.ac.uk

## ABSTRACT

**Introduction**  The number of people living with multiple long-term conditions (MLTCs) is predicted to rise. Within this population, those also living with frailty are particularly vulnerable to poor outcomes, including decreased function. Increased physical activity, including exercise, has the potential to improve function in those living with both MLTCs and frailty but, to date, the focus has remained on older people and may not reflect outcomes for the growing number of younger people living with MLTCs and frailty. For those with higher burdens of frailty and MLTCs, physical activity may be challenging. Tailoring physical activity in response to symptoms and periods of ill-health, involving family and reducing sedentary behaviour may be important in this population. How the tailoring of interventions has been approached within existing studies is currently unclear. This scoping review aims to map the available evidence regarding these interventions in people living with both frailty and MLTCs.

**Methods and analysis**  We will use a six-stage process: (1) identifying the research questions; (2) identifying relevant studies (via database searches); (3) selecting studies; (4) charting the data; (5) collating and summarising and (6) stakeholder consultation. Studies will be critically appraised using the Mixed Methods Appraisal Tool.

**Ethics and dissemination**  All data in this project will be gathered through database searches. Stakeholder consultation will be undertaken with an established patient and public involvement group. We will disseminate our findings via social media, publication and engagement meetings.

## Strengths and limitations of this study

⇒ We will identify primary studies that have incorporated sedentary behaviour, physical activity and/or exercise interventions and report how they were modified and tailored to the needs of those with multiple long-term conditions and frailty.
⇒ The design of the review has been shaped by strong patient and public involvement, and a diverse range of stakeholders will be involved throughout the review.
⇒ The exclusion of non-English language studies may represent a limitation.
⇒ There is potential that our review may miss some relevant studies, but an initial search indicates a need to maintain a balance between reviewing a feasible number of studies and ensuring all relevant articles are included.

## INTRODUCTION
### Physical inactivity and sedentary behaviour in people with multiple long-term conditions and frailty

The number of people living with multiple long-term conditions (MLTCs), defined as the coexistence of two or more chronic conditions, is a growing global health concern.[1] The presence of MLTC is associated with a myriad of poor outcomes, including low physical function and a resulting loss of independence and life participation.[2–9] Within this population, those also living with frailty, defined as 'a multidimensional syndrome of decreased physiological reserve leading to increased vulnerability to minor health stressors', represent a group who are particularly vulnerable to these poor outcomes.[10 11] This is, in part, due to a bi-directional relationship between frailty and MLTCs.[12 13] Frailty appears to contribute to the onset of MLTCs, while the presence of MLTCs increases the risk of developing frailty.[14 15] Evidence suggests that people with MLTCs are at risk of becoming frail earlier.[16 17]

Sedentary behaviour (defined as any activity performed in a seated or reclined posture that requires low energy expenditure) and physical inactivity (insufficient amounts of moderate-to-vigorous physical activity) are associated with many long-term conditions.[18–20] Increasing physical activity and encouraging people to participate in exercise are often primary targets for intervention, but there is also growing interest in reducing sedentary behaviour.[21] Physical inactivity is also common within the context of MLTCs. Approximately 29% of people living with MLTCs are unable to achieve physical activity recommendations of at 150 min of moderate physical activity per week, with levels of inactivity increasing with age.[22] In addition, there is an increased prevalence of probable sarcopenia in people with MLTCs.[23] Within this group there is a graded relationship between the degree of multimorbidity and sarcopenia, and an independent association between sarcopenia and physical inactivity.[24] People with MLTCs who can exercise demonstrate improved outcomes, including increased function, quality of life and life expectancy, indicating that physical activity and exercise may also be important in the management of MLTCs.[21 22 25–29]

Sarcopenia, physical inactivity and sedentary behaviour are also modifiable characteristics of the frailty phenotype, and existing evidence suggests that being physically active, including undertaking resistance training, may delay both the onset and progression of frailty and improve outcomes such as physical function.[30–35] Increased physical activity is also a cornerstone of obesity management, which has also been associated with both MLTCs and frailty.[36–38] Reducing sedentary behaviour and engaging in physical activity and exercise, across the life-course, and when living with MLTCs, is now a major focus of clinical and public health policy.[21 32 39–41]

Compared with data from 2015, an 86% increase in the number of people living with two more conditions is predicted by 2035, in those aged over 65 years alone, making effective interventions for this groups increasingly important.[42 43] This is underscored by a recent Public Health England report which underlines the negative impact of the COVID-19 pandemic on sedentary behaviour and physical activity, particularly on those with MLTCs who were advised to shield. Modelling predicts that the impact of deconditioning and increased fall risk will lead to an estimated cost of £211 million per year to health and social care services.[44]

## Sedentary behaviour and physical activity and interventions

Existing systematic reviews indicate that structured exercise can lead to improvements in function, depression and anxiety, but have mixed impacts on quality of life in people living with MLTCs who are not frail.[45–47] Similarly, systematic reviews of exercise in people living with frailty suggest that those which include resistance training may improve function but have an unclear impact on quality of life or ability to participate in activities of daily living.[48 49] Only one of these reviews, published in 2012, has examined the effectiveness of exercise for people living with

both MLTCs and frailty, but focuses specifically on older people (aged over 60 years), and is limited by the interchangeable use of the concepts of disability and frailty.[45] This review indicated that exercise had a beneficial effect on mobility and physical functioning in this group, but not on quality of life. Due to the range of intervention types included, the authors were unable to determine the most effective type of programme, and there was little discussion of any adaptations made, except for the phased approach in which supervision was gradually reduced in one study. Contemporary research indicates that the demographics of people living with frailty and MLTCs are changing, with accelerated rates of both phenomena being observed in younger populations, people living with obesity, those with new and emerging health issues such as long-term HIV and early onset type II diabetes.[15 36 50 51] An updated review of sedentary behaviour and physical activity interventions for those living with both MLTCs and frailty that includes these groups is warranted.[52]

Existing evidence suggests that identifying and targeting non-disease specific issues (eg, symptoms such as pain, fatigue, breathlessness) are important factors associated with increased intervention effectiveness.[47 53] This may be important within the context of physical activity and exercise interventions, where uptake is hampered by concerns that symptoms may be aggravated.[54] Additionally, adherence in this population is low and the sustained engagement of people with both MLTCs and frailty is disrupted by periods of ill-health, exacerbation of their conditions and increased symptom burden, coupled with lack of guidance and support on safe and effective physical activity in these circumstances.[36 55–57]

Guidance on the management of people living with MLTCs and frailty highlights a need to identify and tailor treatment strategies.[41] Tailored adaptations to physical activity and exercise may encourage people living with frailty and MLTC to engage and support them to continue as much as they are safely able during periods of ill-health, increased symptomology and functional variability. Such strategies are urgently needed, as those who do complete combined resistance and aerobic exercise programmes appear to benefit from a range of improved outcomes, including improved exercise performance, physical activity levels and health status.[55] While recently published guidance supports healthcare professionals to address concerns relating to the exacerbation of common symptoms at the point of initiating physical activity and exercise, they do not address how such interventions might need to be modified, which appears to be important for sustainability.[54] Even less is known about the role of interventions to address sedentary behaviour in this population.[20 53 58–60] Reducing sedentary behaviour may be a useful adjunct approach for those who lack the capacity or motivation to undertake physical activity and exercise. For these individuals, reducing and breaking up sedentary time may offer a more realistic, acceptable starting point, and a more sustainable segue into exercise.[61 62]

Finally, contextual factors, such as family involvement have been cited as key factors which may influence engagement and consequently outcomes.[47] Indeed, qualitative research indicates that family members and significant others are important 'gatekeepers' to health services and independence in people living with MLTCs and frailty.[58 63] This suggests that family involvement may be an important component of an intervention. Evidence also suggests that MLTCs may occur within families beyond parent/child dyads, particularly within people from minority ethnic backgrounds.[64–67] These are associated with increased sedentary behaviour, physical inactivity and poor levels of function at the family level.[64–69] Therefore, in addition to improving the engagement of those living with MLTCs and frailty, the involvement of family members may help to reduce their risk of developing MLTCs or support them to manage existing LTCs. Indeed, the findings of a recent systematic review suggest that the engagement of carers in 'dyad' during structured exercise leads to improvements in both caregiver physical and psychosocial health.[70] The involvement of carers in sedentary behaviour and physical activity interventions is less clear. Caregiving requirements are likely to vary by age, and as a result of different clusters of MLTCs, together with frailty, so different caregivers and family members may need to be involved in different ways. Clarifying how carers and family members have been involved in sedentary behaviour, physical activity and exercise interventions for people with MLTCs and frailty may help identify further ways to tailor such interventions and leverage intergenerational support.

Overall, understanding the characteristics of sedentary behaviour, physical activity and exercise interventions for people living with both MLTCs and frailty is an important first step to identifying safe, scalable and sustainable physical activity interventions which may be beneficial in this population.

### Justification for this review

Scoping reviews seek to 'determine the body of literature on a given topic and give a clear indication of the volume of literature available, as well as an overview of its focus'.[71] While there is emerging evidence in the context of MLTCs and frailty independently, less is known about the role of sedentary behaviour and physical activity interventions in those living with both. Specifically, we would like to identify primary studies in these conditions, including younger adults where very little research has been conducted. Conducting a scoping review will enable us to draw on the wider literature and begin to develop a programme theory to underpin an intervention targeting this increasing population.[72–76] Of relevance to this population is scoping the range of ways in which interventions may be adapted and tailored. An approach that allows diverse evidence to be integrated will provide a more nuanced understanding of this, which may be lost within a systematic review.[76 77]

### Aim and objectives

The aim of this review is to map the available evidence on the use of sedentary behaviour, physical activity and exercise interventions in people living with both frailty and MLTCs. The identified evidence will be used to highlight gaps within the existing literature, including a determination of whether there is sufficient evidence in this area to undertake a systematic review. The results of the review will be used to inform the design and development of an intervention for this population.[73]

## METHODS AND ANALYSIS
### Design

This scoping review will follow a six-stage process informed by guidance from Arksey and O'Malley,[78] and subsequent refinements outlined by Levac et al,[79] Colquhorn et al[80] and Daudt et al.[81] This protocol follows the Preferred Reporting Items for Systematic Reviews and Meta-Analyses (PRISMA) reporting guidelines for scoping reviews.[82]

### Identifying the research questions

To achieve the above aims the following questions will be addressed:
- What are the characteristics of sedentary behaviour, physical activity and exercise interventions that have been used with people living with both frailty and MLTCs?
- How have carers and relatives been included within the design, development and delivery of these interventions?
- For each of the above, what approaches appear to contribute to improved engagement and outcomes, particularly physical function?

Patient and public representatives have been involved in shaping these questions from the outset. They described struggling to maintain physical activity levels during periods of ill-health, compounded by barriers to accessing existing services which could not accommodate their needs, or their variable ability to engage .

### Identifying relevant studies
#### Inclusion and exclusion criteria

All relevant literature will be included, irrespective of the study design used. Studies will be included if the meet the following inclusion criteria (summarised within table 1). These parameters may be refined and adapted if unmanageable volumes of eligible studies are identified following an initial search.[78]

#### *Population*

Adults aged 18 years or above, living with both MLTCs and frailty. Within the review, we define MLTCs as the co-existence of two or more chronic conditions (physical or mental) in a single individual.[1] Included long-term conditions, which could feasibly be influenced by physical activity, are outlined in table 2. Studies where the presence of MLTCs could be assumed based solely on participants'

**Table 1** Inclusion and exclusion criteria

| Inclusion criteria | Exclusion criteria |
|---|---|
| Any study design examining interventions or intervention content (eg, single-group preintervention and postintervention, parallel-group, crossover or cluster designs, qualitative studies and process evaluations relating to interventions) | Studies in children or animals |
| Adults aged 18 years and above | Presence of MLTCs not defined or <50% of the sample report MLTCs |
| Living with both frailty and MLTCs | Recognised measure of frailty or validated proxy not used |
| Interventions with a sedentary behaviour, physical activity or exercise focus, including multicomponent interventions | Non-English language studies |
| Any setting | Studies of acute responses to sedentary behaviour, physical activity or exercise, including interventions of <1 week in duration |
| MLTC, multiple long-term condition. | |

age or circumstances (eg, resident of a nursing home) will be excluded.[53] However, given the relatively recent use of MLTCs as a term, we will include studies where the characteristics of the sample indicate that the majority (over 50%) are living with MLTCs (eg, Charlson Comorbidity Index scores, counts of conditions). Similar approaches have been adopted within previous reviews of exercise and MLTCs.[46]

We define frailty as 'a multidimensional syndrome of decreased physiological reserve leading to increased vulnerability to minor health stressors'.[10] Studies using a validated frailty measure (eg, Fried Frailty Phenotype, Clinical Frailty Scale, Frailty Index, Electronic Frailty Index) will be prioritised. Considering that frailty in younger people with MLTCs has only recently begun to be recognised, we will also include studies using proxy indicators of frailty (outlined within table 3).[17 83–88] Studies which use other measures and cite supporting evidence to suggest association with frailty will also be included. Accepting that these instruments have only

moderate specificity for the identification of frailty, and have yet to be validated within younger frail populations, they may be more appropriate to, and more commonly used in, younger groups.[17 85] Studies including participants described as frail with no quantitative measurement of this will be excluded.

**Table 2** Conditions included within the review

| | |
|---|---|
| Type 2 diabetes | Asthma |
| COPD | Arthritis (osteo and rheumatoid) |
| Depression | Anxiety |
| Cancer (solid organ, haematological and metastatic) | HIV and AIDS |
| Chronic kidney disease | Chronic liver disease |
| Heart failure | Peripheral artery disease |
| Coronary artery disease | Hyperlipidaemia |
| Obesity | Ischaemic heart disease |
| Osteoporosis | Multiple sclerosis |
| Parkinson's disease | |
| AIDS, Acquired immunodeficiency syndrome ; COPD, chronic obstructive pulmonary disease; HIV, Human Immunodeficiency virus. | |

**Table 3** Functional measures which are recognised proxy measures of frailty, and published cut-offs indicative of frailty

| Function test | Published cut points for the identification of frailty |
|---|---|
| Modified physical performance test | Score range: 0–36: <br> 1. Not frail: 32–36 <br> 2. Mild frailty: 25–32 <br> 3. Moderate frailty: 17–24 <br> 4. Dependent: <17 |
| Balance performance oriented mobility assessment (BPOMA) | BPOMA >19 |
| Short physical performance battery | A score of ≤7 is indicative of frailty |
| Timed get-up-and-go test | A score of ≥9 s |
| Gait speed test | A gait speed of 0.8 m/s is indicative of frailty <br> Taking >5 s to walk 4 m |
| Sit to stand tests: <br> 1. Sit to stand 10 s <br> 2. Sit to stand 30 s <br> 3. Sit to stand 60 s <br> 4. Sit to stand 5 repetitions <br> 5. Sit to stand 10 repetitions | Dependent on the type of sit to stand test used: <br> 1. ≥10 s for the five times sit stand <br> 2. Lower than published criterion standards to maintain independence, stratified by age and gender for other forms of the test |
| Handgrip strength | Scores within the lowest quartile, stratified by sex |
| Strength, assistance with walking, rising from a chair, climbing stairs, and falls (SARC-F) questionnaire | Score of ≥4 |

Studies which target only carers will be included, providing they are delivering unpaid care for people with the above inclusion criteria and the interventions focus on sedentary behaviour, physical activity or exercise. Carer involvement is not included as a specific inclusion criterion, as we do not want to exclude relevant studies which did not include carers.

### Concept
We define physical activity as 'people moving, acting and performing within culturally specific spaces and contexts, and influenced by a unique array of interests, emotions, ideas, instructions and relationships'.[89] Within this, we include exercise interventions as 'planned, structured and repetitive bodily movement with the objective of improving or maintaining physical fitness'.[90] Multicomponent interventions, including rehabilitation programmes, will also be eligible, provided they meet the other inclusion criteria described. We will also include interventions which target sedentary behaviour, as previously defined.[18] Studies examining the acute effects of physical activity or sedentary behaviour interventions are excluded, as we are interested in the characteristics of programmes which are designed to influence long-term outcomes such as physical function. An intervention duration of 1 week distinguishes acute interventions from non-acute behavioural interventions in free-living conditions.[91]

### Context/Settings
Studies from all settings will be eligible, including those in community, workplace, residential and hospital settings.

Searches will be limited to the year 2000 onwards to ensure they are relevant to current practice, and because the term 'frailty' as a syndrome of increased vulnerability is not well recognised prior to this.[92] Non-English language studies will be excluded. Any study design examining interventions or intervention content will be eligible for inclusion. The reference lists of relevant reviews will be used to identify other eligible papers not already included within the review. This will allow us to exclude papers which do not meet the eligibility criteria, and to avoid overstating the results of papers already included as primary sources.

### Information sources
We will search for studies using the following databases and trials registries:
► For systematic reviews: Cochrane; PROSPERO; Database of Abstracts of Reviews of Effects.
► For published research: MEDLINE; EMBASE; Cumulative Index to Nursing and Allied Health (CINAHL); Web of Science; Sports Discus; PsycINFO; Pedro; Allied and Complementary Medicine; Cochrane Central Register of Controlled Trials (CENTRAL); Scopus.
► For grey literature: internet searching (eg, Google Scholar), BIOSIS previews, Open Grey and the Index to Scientific and Technical Proceedings.

Relevant ongoing clinical trials will be included if they provide sufficient information. The following databases will be searched for ongoing trials: CENTRAL; US National Institutes of Health Ongoing Trials Register (ClinicalTrials.gov) and the WHO International Clinical Trials Registry Platform.

Meeting abstracts will not be included as they are unlikely to include the level of information required. Additional relevant literature will be identified via hand-searching the references of included papers, drawing on forward and backward citation tracking and electronic 'cited by' searches using Google Scholar. Where included trials reference linked protocol papers or qualitative research, these will be included. Searches will be updated prior to publication.

### Searches
Search terms used are outlined in online supplemental appendix 1. This strategy has been developed by a health information specialist and the review team for MEDLINE and will be translated for other databases.

Searches will be executed by HMLY with support from the information specialist where required. Initially MEDLINE (PubMed) and CINAHL will be searched to pilot the strategy. Key words from the titles, abstracts and index terms used to describe the retrieved papers will be reviewed. The research team will then meet to discuss any refinements before further searches are conducted.[79] An initial search for terms relating to frailty/function OR terms relating multimorbidity AND terms relating to physical activity/sedentary behaviour/exercise yielded an unmanageable amount of data. Consequently, the search was adjusted to frailty/function terms AND multimorbidity terms AND physical activity/sedentary behaviour/ exercise terms.

### Selection of evidence
Following de-duplication, the remaining studies will be independently screened for eligibility by two reviewers, using the inclusion criteria outlined above. A systematic approach will be facilitated by reference and review management software (EndNote V.X9 and Covidence).[79 81] First, titles and abstracts will be screened to remove ineligible records. Following this, at least two reviewers will screen the remaining full-text copies. At this stage, demographic and clinical characteristics of the studies will be assessed to determine if they meet the criteria for MLTCs and frailty. Any discrepancies will be resolved through discussion and the inclusion of an additional reviewer if required. Authors will be contacted if, after full-text screening, it is still unclear whether to include/exclude an article. Authors will be contacted via email, with a further follow-up email 2 weeks later. The review team will meet regularly throughout each stage.

### Data charting
A standardised data charting form will be used to comprehensively extract data from the included studies. The

form will be developed by the review team based on the key objectives of the review. It will be piloted by two reviewers on five studies to ensure all relevant information is extracted. Any changes to the form, and the rationale for these, will be recorded.

Microsoft Excel will be used to manage the extracted data. Where key information is missing from the full texts, authors will be contacted for additional information. The extraction of intervention characteristics, including how they are tailored and adapted will be guided by the template for intervention description and replication (TIDieR) checklist and the Consensus on Exercise Reporting Template (CERT).[93 94] For data relating to study methodology, relevant reporting guidelines will be used to guide the data extracted for each study type.[76] Key data to be extracted is outlined within table 4. This list is not exhaustive, and data extracted may be subject to refinement.[76]

To differentiate between physical activity and sedentary behaviour interventions, we will use the approach described by Hadgraft et al.[91] Finally, we will record the outcomes used to determine the interventions effectiveness, including outcomes relating to carer health and well-being, and the effects of the intervention on physical function and other relevant outcomes (particularly physical activity, sedentary behaviours and measures of quality of life). Key findings that relate to the scoping review questions will also be recorded alongside an assessment of the quality of the study.

### Critical appraisal

Critical appraisal of the included studies will provide information on the quality of the available evidence. The identification of lower quality research will strengthen the identification of gaps within the existing literature.[79] Studies will not be excluded based on quality.

We will assess the quality of the included studies using the using the Mixed Methods Appraisal Tool (MMAT), which has previously been used in scoping reviews.[95–97] The MMAT is a brief, but reliable, critical appraisal tool which provides a single method for assessing the quality of a range of qualitative, quantitative and mixed-methods study designs.[76 98 99] Two reviewers will be familiarised with the tool and devise a strategy for applying the tool in a systematic manner.[98 99] Following this, they will undertake the assessment independently, with recourse to additional members of the team where required.[99] A detailed presentation of the ratings of each included criterion will be reported.[100]

### Synthesis and reporting

We plan to use a convergent synthesis design, based on each of the research questions identified.[101] Qualitative and quantitative data will be summarised using a narrative approach, supplemented by descriptive statistics, tables and figures as appropriate. Following this, qualitative and quantitative data relating to each question will be integrated using mixed-methods joint displays.[101] We

anticipate that joint displays will be organised according to the domains outlined within the TIDieR and CERT checklists.[93 94] Within the joint displays, consideration will also be given to the effectiveness of the identified interventions and the quality of the studies. This will enable us to better identify those interventions or components which appear to lead to more favourable outcomes, and to assess areas of ongoing uncertainty. The results of this review will be reported in accordance with PRISMA guidance.[82] This proposed synthesis plan will be further refined towards the end of the review.[76]

### Stakeholder consultation

The final stage will involve discussing the review results with two stakeholder groups. The first will comprise approximately six to eight people living with both frailty and at least two co-existing long-term conditions from those included within this review, alongside their carers/family members. These individuals are already engaged as members of a patient and public involvement (PPI) group for a study related to this review. We have already taken steps to ensure that we are actively considering equality, diversity and inclusion and will continue to ensure the composition of this group is broadly representative of the population of interest.

The second group will comprise approximately six to eight exercise and healthcare professionals, and researchers with expertise in intervention development. These individuals are already engaged as collaborators for a study related to this review or will be approached via existing links with local hospital and community health and research networks. We will include broad representation from professionals and academics with an interest in the management of MLTC and frailty alongside expertise in the specialty areas of interest in the review (respiratory, musculoskeletal, mental health, metabolic and infectious diseases, cardiology, neurology oncology and geriatrics).

The two groups will meet separately to mitigate any issues relating to power differentials, and to allow both groups to discuss their views openly. A single meeting for each group will be facilitated by HMLY. Meetings will not be audio-recorded but a co-facilitator will take notes and observe group interactions. Meetings will be held face-to-face or virtually as circumstances allow. Given the heavy burden of healthcare and the increased potential for periods of illness, where lay stakeholders cannot attend, they will be consulted individually. Face-to-face meetings will be held in accessible locations. Materials will be adapted for the needs of those with sensory impairments as required. Lay members will be rewarded and recognised for their time and expertise in accordance with current guidance.[102]

The objective of the meetings will be to understand stakeholders' perspectives regarding any potential evidence informed interventions identified.[73] We will present a summary of our findings to inform discussions. They will be asked to consider how well any identified interventions fit the proposed context, and where and

**Table 4** Key data to be extracted within the review

| | |
|---|---|
| Study details | Author(s) |
| | Type of publication |
| | Year of publication |
| | Country of origin |
| Description of methodology | Aims/Purpose |
| | Study design |
| | Inclusion and exclusion criteria |
| | Definition of frailty and frailty assessment or proxy functional measure used |
| | Primary and secondary outcomes |
| | Where applicable, definition of the carers involved |
| | Setting/Context (geographical, cultural, social environment and the organisational and political systems in which an intervention occurs) |
| | Sample size |
| | Characteristics of the study population, including:<br>Ethnicity<br>1. Age<br>2. Sex<br>3. Indicators of socioeconomic status<br>4. Presence of cognitive impairment<br>5. Number, type and severity of long-term conditions<br>6. Level of frailty of participants |
| | Where applicable, characteristics of carers or family members or significant others, including relationship to care receiver |
| Description of intervention | Focus of the intervention (sedentary behaviour, physical activity, exercise or combination) |
| | If applicable, the type of physical activity/ exercise, including equipment used and an outline of the components included |
| | The methods used to prescribe the intervention |
| | The decision rules for determining the starting level |
| | The intervention duration and dose, that is, the prescribed frequency and intensity, the duration of the intervention and any maintenance period |
| | The mode of delivery (face-to-face, virtual, individual or group) |
| | The decision rules for determining progression |
| | Details of how the programme was progressed and how this was monitored |
| | The location/setting of delivery (eg, home-based or in-centre) including any necessary infrastructure or other relevant features |
| | Details, methods of and reasons for tailoring, personalisation or adaptation. |
| | Details and methods of any modifications to the intervention during the study, particularly in relation to periods of ill-health and fluctuating symptomology |
| | Intervention rationale, programme theory or goals |
| | The physical or informational materials used in the intervention, including those provided to participants or used in intervention delivery or in training of intervention providers |
| | The procedures, activities and/or processes used in the intervention, including any enabling or supportive activities, motivation strategies used (eg, counselling/education; environmental modification; prompting; self-monitoring; social comparison; financial incentives) |
| | Amount of supervision, including contact time |
| | The intervention providers, including their qualifications/expertise, background and any training provided to them |
| | Description of how carers, relatives or significant others are included within the design, development or delivery of the intervention |
| | How intervention fidelity was assessed, and by whom, including methods for measuring adherence will also be included |
| | Strategies used to maintain or improve fidelity |
| | How well the intervention was delivered as planned, including recorded levels of adherence to the programme |

how they may need further adaptation.[74] Areas where there are ongoing uncertainties will be flagged. Stakeholders will also be asked for their views on areas which have not been addressed within the current evidence base which may be important to them. Following the meetings, a plan for future development and/or adaptation work will be developed and shared with the whole group for further comment. Feedback from the stakeholder consultation will be integrated with the findings of the review and described in the final report.

### Patient and public involvement

In addition to shaping the research questions, PPI group members have provided feedback on how to make stakeholder meetings inclusive and accessible. They have also advised on the dissemination strategy adopted.

### ETHICS AND DISSEMINATION

All data in this project will be gathered through database searches. Ethical approval to publish information from the stakeholder consultation process will be sought from the University of Leicester Internal Review Board for the stakeholder consultation stage of this review.

We will disseminate our findings at relevant academic and clinical meetings and by publishing them in an academic journal within the field. Results will also be disseminated to people living with both frailty and MLTC, and carers via the Lifestyle and Cardiovascular Biomedical Research Units involvement forum, relevant local and national patient charities, and our social media platforms.

### SUMMARY

The numbers of people living with MLTC are predicted to rise, and within this population those also living with frailty are particularly vulnerable to poor outcomes such as decreased function and independence. Sedentary behaviour, physical activity and exercise interventions form an integral part of chronic disease and frailty management and may also be important for those living with both MLTC and frailty. Existing systematic reviews suggest that exercise has the potential to improve function in those living with both MLTCs and frailty, but to date the focus has remained on older people, which may not reflect those living with MLTC, who become frail at a younger age. Tailoring the intervention in response to symptoms and periods of ill-health and involving family members appears to be important in this population, but how this has been approached within existing studies is currently unknown. Additionally, the role of broader interventions which address sedentary behaviour are unclear. This scoping review aims to map the available evidence on the use of sedentary behaviour, physical activity and exercise interventions in people living with frailty and MLTCs. The results will inform the design of a tailored intervention and highlight gaps, directing future research.

**Author affiliations**
[1]Leicester Diabetes Centre, University Hospitals of Leicester NHS Trust, Leicester, UK
[2]Diabetes Research Centre, College of Life Sciences, University of Leicester, Leicester, UK
[3]MRC Epidemiology Unit, Institute of Metabolic Science, Cambridge Biomedical Campus, Cambridge, UK
[4]Baker Heart and Diabetes Institute, Melbourne, Victoria, Australia
[5]NIHR Applied Research Collaboration East Midlands, University Hospitals of Leicester NHS Trust, Leicester, UK
[6]NIHR Leicester BRC, University Hospitals of Leicester NHS Trust, Leicester, UK
[7]Medical School, University of Edinburgh, Edinburgh, UK
[8]Library Research Services, University of Leicester, Leicester, UK
[9]Department of Respiratory Medicine, University Hospitals of Leicester NHS Trust, Leicester, UK
[10]Centre for Exercise and Rehabilitation Science, Leicester Biomedical Research Unit, Leicester, UK

**Contributors** HMLY conceived the study idea. HMLY, PCD, LYH, JS, JH, FC, HS, PJH, SL, MH, RP, SJS, TY, MJD designed the protocol. HMLY, MH and RP devised the stakeholder collaboration process. HMLY and SL designed the search strategy. HMLY prepared the manuscript. PCD, LYH, JS, JH, FC, HS, PJH, SL, MH, RP, SJS, TY, MJD reviewed final manuscript. Mentorship was provided by TY, SJS, MJD. HMLY, PCD, LYH, JS, JH, FC, HS, PJH, SL, MH, RP, SJS, TY, MJD contributed important intellectual content during manuscript drafting or revision. HMLY accepts accountability for the overall work ensuring that questions pertaining to the accuracy or integrity of any portion of the work are appropriately investigated and resolved.

**Funding** The research was funded by the National Institute for Health Research (NIHR) Leicester Biomedical Research Centre, Collaboration for Leadership in Applied Health Research and Care East Midlands (CLAHRC EM). HMLY is supported by grants from the NIHR (NIHR301593). SJS is supported by the Collaboration for Leadership in Applied Health Research and Care East Midlands.

**Disclaimer** The views expressed in this publication are those of the authors and not necessarily those of the NHS, the NIHR or the Department of Health and Social Care.

**Competing interests** None declared.

**Patient and public involvement** Patients and/or the public were involved in the design, or conduct, or reporting, or dissemination plans of this research. Refer to the 'Methods and analysis' section for further details.

**Patient consent for publication** Not applicable.

**Ethics approval** Not applicable.

**Provenance and peer review** Not commissioned; externally peer reviewed.

**ORCID iDs**
Hannah M L Young http://orcid.org/0000-0002-4249-9060
Rebecca Pritchard http://orcid.org/0000-0002-2655-7312

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
