## [Reviewer comments · BMJ Open]

ARTICLE DETAILS

TITLE (PROVISIONAL)	Physical activity and sedentary behaviour interventions for people living with both frailty and multiple long-term conditions: a scoping review protocol
AUTHORS	Young, Hannah; Yates, Thomas; Dempsey, Paddy; Herring, Louisa; Henson, Joseph; Sargeant, Jack; Curtis, Ffion; Sathanapally, Harini; Highton, Patrick; Hadjiconstantinou, Michelle; Pritchard, Rebecca; Lock, Selina; Singh, Sally; Davies, Melanie

VERSION 1 – REVIEW

REVIEWER	Trott, Mike Anglia Ruskin University - Cambridge Campus
REVIEW RETURNED	09-Feb-2022

GENERAL COMMENTS	This protocol is a well written and clearly thought out review, and overall I congratulate the authors on a job well done. I have several very minor comments which I will outline below, and two comments that I suppose are more than just the odd typo and comment: 1. How will you deal with reviews in the both the data collection and the evidence synthesis? For example, you will likely come across reviews and the primary papers that are included in said reviews - so how will that be managed? Also, if you do find reviews, the papers included in these reviews might be pre-2000, and therefore strictly speaking would not fit into your inclusion criteria.2. I see that you have excluded studies that are not written in English. This is a pretty big limitation, especially given that it is perfectly possible that not many studies will be found in the search. Is it possible you could recruit someone into the review team that is able to translate papers for you? Other comments: Introduction Line 38: the expression 'are hallmarks' is fairly ambiguous, can you please be a little more specific as to what is meant? Lines 38-19: the term Reducing physical inactivity is a double negative, which makes it harder to read.
--

	Line 45: physical activity recommendations - I assume you mean by this the UK department of health guidelines? Can you please elaborate on this and state exactly what the guidelines are? Line 53: 'People with MLTCs who can exercise demonstrate improved outcomes.' Can you add one or two of these outcomes to give more context? The same with Line 7 on the next page. Line 32: the last sentence before the sedentary behaviour section: I assume you mean the cost is 211m per year to UK health and social care services? Line 49: 'published almost ten years ago' - please be specific with dates here. Although now, at this time, this statement will be true, in 5 years time it will not. Line 53: can you elaborate on the results of this review 'that was published more than ten years ago?' Perhaps the results of this can be used to formulate a hypothesis or two? Page 12 line 35 typo - 'the' (i think it should be 'they') Table 1: I am glad to see that you are excluding studies of interventions of less than one week, and I were doing this review I would do the same thing! I think, however, some justification is needed as to why you are choosing to exclude these studies, as without the justification is sort of comes out of nowhere. Data charting: line 44: can you be more specific about contacting the authors for additional information? How long will you give them until you exclude? Will you try more than once? One more thing to consider in general (not related to the manuscript as it stands) is the issue of adherence. Indeed, we know that decreasing SB and increasing exercise and PA yield significant benefits, however the adherence of these interventions is often not reported (mainly because they are not measured), and is a very important thing to discuss in the review as it really is one of the biggest issues with PA related interventional studies. Again, congratulations on a well written review protocol. I look forward to the review itself.
--	---

REVIEWER	MacDermid, Joy University of Western Ontario, Physical Therapy
REVIEW RETURNED	21-Feb-2022

GENERAL COMMENTS	well described protocol; clear , methods appropriate, enough detail without extra verbage
---

VERSION 1 – AUTHOR RESPONSE

Reviewer 1

1. How will you deal with reviews in the both the data collection and the evidence synthesis? For example, you will likely come across reviews and the primary papers that are included in said reviews - so how will that be managed? Also, if you do find reviews, the papers included in these reviews might be pre-2000, and therefore strictly speaking would not fit into your inclusion criteria.

This is a good point. We have updated the manuscript to detail how we will deal with this eventuality as follows:

“Any study design examining interventions or intervention content will be eligible for inclusion. The reference lists of relevant reviews will be used to identify other eligible papers not already included within the review. This will allow us to exclude papers which do not meet the eligibility criteria, and to avoid overstating the results of results of papers already included a primary sources.”

2. I see that you have excluded studies that are not written in English. This is a pretty big limitation, especially given that it is perfectly possible that not many studies will be found in the search. Is it possible you could recruit someone into the review team that is able to translate papers for you?

This is an important point, and one that we did consider when devising our scoping review protocol. Unfortunately, we do not have the language expertise within our team to translate non-English language papers. Although the inclusion of a professional translator would be optimal, we have identified 17,444 records for title and abstract review, so the resource needed to review non-English studies may be significant, and it is difficult to determine how many languages we would need to translate at this point in the review.

Unfortunately, we do not have the resources required to employ professional translators. We acknowledge this as a limitation in the protocol and will also do so in the final review. Interestingly, two reviews (referenced below) suggest that the effect of restricting systematic reviews to English-language publications is negligible, although we appreciate that it may be more impactful for a scoping review, where the aims is to map the available evidence.

- Dobrescu AI, Nussbaumer-Streit B, Klerings I, Wagner G, Persad E, Sommer I, Herkner H, Gartlehner G. Restricting evidence syntheses of interventions to English-language publications is a viable methodological shortcut for most medical topics: a systematic review. *Journal of Clinical Epidemiology*. 2021 Sep 1;137:209-17.
- Morrison A, Polisena J, Husereau D, Moulton K, Clark M, Fiander M, Mierzwinski-Urban M, Clifford T, Hutton B, Rabb D. The effect of English-language restriction on systematic review-based meta-analyses: a systematic review of empirical studies. *International journal of technology assessment in health care*. 2012 Apr;28(2):138-44.

Line 38: the expression 'are hallmarks' is fairly ambiguous, can you please be a little more specific as to what is meant?

Thank you. We have tried to be more specific by amending to:

“...physical inactivity (insufficient amounts of moderate-to-vigorous physical activity) are associated with many long-term conditions”.

Lines 38-19: the term Reducing physical inactivity is a double negative, which makes it harder to read.

Many thanks for spotting this, we have now amended to 'increasing physical activity'.

Line 45: physical activity recommendations - I assume you mean by this the UK department of health guidelines? Can you please elaborate on this and state exactly what the guidelines are?

The authors of the paper referenced used the recommended physical activity levels of 150 min of moderate physical activity per week, taken from US physical activity recommendations from 2008. These are now outdated, but still appear to reflect current UK guidance from the Chief Medical Officers, so we have updated to make this clearer as follows:

“Physical inactivity is common within the context of MLTCs. Approximately 29% of people living with MLTCs are unable to achieve physical activity recommendations of at 150 minutes of moderate physical activity per week, with levels of inactivity increasing with age.”

Line 53: 'People with MLTCs who can exercise demonstrate improved outcomes.' Can you add one or two of these outcomes to give more context? The same with Line 7 on the next page.

Absolutely. We have added further detail as follows to the two sections highlighted:

“People with MLTCs who can exercise demonstrate improved outcomes, including increased function, quality of life and life expectancy, indicating that physical activity and exercise may be important in the management of MLTCs.”

And

“...the frailty phenotype and existing evidence suggests that undertaking resistance exercise and physical activity may delay both the onset and progression of frailty and improve outcomes such as physical function.”

Line 32: the last sentence before the sedentary behaviour section: I assume you mean the cost is 211m per year to UK health and social care services?

That is correct, we have deleted the space between the sum and the million to hopefully make this clearer.

Line 49: 'published almost ten years ago' - please be specific with dates here. Although now, at this time, this statement will be true, in 5 years' time it will not.

This is a good point. We have updated to reflect the year published.

Line 53: can you elaborate on the results of this review 'that was published more than ten years ago?' Perhaps the results of this can be used to formulate a hypothesis or two?

We have added some further information as follows:

“This review indicated that exercise had a beneficial effect on mobility and physical functioning in this group, but not upon quality of life. Due to the range of intervention types included the authors were unable to determine the most effective type of programme, and there was little discussion of any adaptations made, except for the phased approach in which supervision was gradually reduced in one study.”

Unfortunately, there wasn't enough information to formulate a hypothesis but adding this detail does serve to underline how little we know about the tailoring of exercise for this group.

Page 12 line 35 typo - 'the' (I think it should be 'they')

We have now corrected this, thank you for highlighting it.

Table 1: I am glad to see that you are excluding studies of interventions of less than one week, and I were doing this review I would do the same thing! I think, however, some justification is needed as to why you are choosing to exclude these studies, as without the justification is sort of comes out of nowhere.

Thank you for highlighting this, we have added further justification as follows on page 14:
“Studies examining the acute effects of physical activity or sedentary behaviour interventions are excluded, as we are interested in the characteristics of programmes which are designed to influence longer-term outcomes such as physical function. An intervention duration of one week distinguishes acute interventions from non-acute behavioural interventions in free-living conditions”.

Data charting: line 44: can you be more specific about contacting the authors for additional information? How long will you give them until you exclude? Will you try more than once?

We have added further information as follows:
“Authors will be contacted via email, with a further follow-up email two weeks later.”

One more thing to consider in general (not related to the manuscript as it stands) is the issue of adherence. Indeed, we know that decreasing SB and increasing exercise and PA yield significant benefits, however the adherence of these interventions is often not reported (mainly because they are not measured) and is a very important thing to discuss in the review as it really is one of the biggest issues with PA related interventional studies.

We agree that this is a key issue to discuss within the review. This information will be captured in the data extraction tool that we will develop. We detail the information we will gather in table 2 as follows:

How intervention fidelity was assessed, and by whom, including methods for measuring adherence will also be included
Strategies used to maintain or improve fidelity
How well the intervention was delivered as planned,

On reflection, the importance of capturing adherence as a measure of how well the intervention was delivered as planned could be emphasised. We have therefore added the following to the final cell of this table:

“...including recorded levels of adherence to the programme.”

We hope that we have adequately addressed each of the reviewers' points and would like to thank them once again for taking the time to read and comment upon our manuscript.

VERSION 2 – REVIEW

REVIEWER	Trott, Mike Anglia Ruskin University - Cambridge Campus
REVIEW RETURNED	28-Mar-2022
GENERAL COMMENTS	Dear authors, Thank you for submitting the revised manuscript and commenting on previous comments made by me and other reviewers. I have nothing to add and recommend this protocol be published.